# Warning people that they are being microtargeted fails to eliminate persuasive advantage
Fabio Carrella [1] ✉, Almog Simchon [2], Matthew Edwards[3] & Stephan Lewandowsky [1,4]

The practice of microtargeting in politics, involving tailoring persuasive messages to individuals based on personal vulnerabilities, has raised manipulation concerns. As microtargeting's persuasive benefits are well-established and its use facilitated by AI tools and personality-inference models, ethical and regulatory concerns are magnified. Here, we explore countering microtargeting effects by creating a warning signal deployed when users encounter personality-tailored political ads. Three studies evaluated the effectiveness of warning "popups" against potential microtargeting by comparing persuasiveness of targeted vs. non-targeted messages with and without popups. Using within subject-designs, Studies 1 ($N = 666$), 2a ($N = 432$), and 2b ($N = 669$) reveal a targeting effect, with targeted ads deemed more persuasive than non-targeted ones. More important, the presence of a warning popup had no meaningful impact on persuasiveness. Overall, across the three studies, personality-targeted ads were significantly more persuasive than non-targeted ones, and this advantage persisted despite warnings. Given the focus on transparency in initiatives like the EU's AI Act, our finding that warnings have little effect has potential policy implications.

Around 62% of the world's population, totaling more than 5 billion people, maintain an active social media profile. The 'typical' social media user now dedicates an average of 2 h and 23 min per day to using these platforms[1]. Consequently, online social network platforms amass an extensive wealth of personal data from users. It has been estimated that just 300 likes on Facebook are sufficient for a machine learning model to better predict an individual's personality than their spouse[2]. Online platforms can even go a step further by predicting data about non-users, often referred to as shadow profiles[3]. These personal data can potentially be utilized by third parties to target advertisements to precisely defined groups of users based on a variety of personal characteristics, such as their sexual orientation or personality.

Targeted advertising can occur without the platform's explicit awareness because advertisers can rely on proxies (i.e., likes and so on) to direct their messages to, say, users of a particular personality type. A well-known case of political microtargeting involved Cambridge Analytica, a media consulting firm that reportedly used surreptitiously collected data from over 50 million Facebook users to target potential voters during the Brexit referendum in the UK with political advertisements[4].

The practice of microtargeting is not entirely new, as profiling and audience segmentation has long been employed in marketing. However, with the advent of digital media, profiling has become more precise as companies gained access to a vast amount of consumer data (e.g., buying habits, sexual tastes, political preferences, and numerous other psychologically-relevant attributes). Psychological microtargeting gives rise to several concerns: first, individuals rarely consent to the use of their personal data, and much of what can be inferred from the data are attributes (e.g., sexual orientation) that people typically wish to keep private. Second, people do not condone microtargeting in general, especially for political purposes[5–8]. Third, microtargeting reinforces a troubling "information asymmetry" wherein platforms and other data practitioners possess extensive knowledge about individuals, whereas people remain unaware of what information these practitioners hold about them[9]. Fourth, if used for political purposes, microtargeting allows for making promises that cannot be subsequently debated or verified by political opponents because messages are only known to sender and receiver. In particular, such promises may be contradictory, with the same party pledging different actions on an issue to each voter, which implies that a subset of those promises was disingenuous. Additionally, political parties might exploit microtargeting to suppress voter turnout among their opponents. For instance, during the 2016 election, the Trump campaign allegedly targeted African-American voters with ads reminding them of Hillary Clinton's past remarks, aiming to discourage their electoral participation[10]. Finally, microtargeting of political ads makes

[1]School of Psychological Science, University of Bristol, Bristol, UK. [2]Department of Psychology, Ben-Gurion University of the Negev, Beer Sheva, Israel. [3]School of Computer Science, University of Bristol, Bristol, UK. [4]Department of Psychology, University of Potsdam, Potsdam, Germany.
✉e-mail: fabio.carrella@bristol.ac.uk

public communication appear as more private, subtle, and less evident as a political message, potentially making people more susceptible and thus acting as a form of manipulation[11,12].

The exact effects of political microtargeting remain subject to debate. Some studies downplay its effects, reporting that political persuasive messages have small effects on individuals [13], or find significant persuasive effects of political messages only when observing specific groups of people or under particularly favorable conditions[14,15]. By contrast, other studies confirm the efficacy of microtargeting, with evidence ranging from the observation that it reinforces party ties and makes voters less likely to defect from their preferred party[16], to showing that personalized political ads tailored to individuals' personalities are more effective than non-personalized ads[17], especially when emotionally charged[18]. Additionally, a systematic review of over 700 experimental studies found that the average message-tailoring effect is around $r = 0.20$ on attitudinal, intentional, and behavioral outcomes[19].

The recent introduction of artificial intelligence tools such as large language models (LLMs) can further boost microtargeting because they enable the automation of message customization, facilitating the large-scale deployment of microtargeting[20]. This process of customization, which used to be painstaking without AI, is now streamlined and rapid.

Modern LLMs can also generate persuasive arguments that enhance message persuasiveness for recipients with diverse linguistic preferences through linguistic feature alignment[21]. Moreover, they can integrate dimensions of social pragmatics, which are fundamental to established psycho-linguistic theories of opinion change[22]. As an example, a recent study found that when participants debated with a version of GPT-4 that had access to personal information, the odds of opinion changes increased by 81.7% compared to when participants debated other humans[23]. Notably, the persuasive advantage dropped significantly when the model did not have access to personal information, although the LLM still outperformed humans in effecting opinion change. Similarly, targeted messages crafted by ChatGPT consistently outperformed non-targeted messages across several studies, irrespective of the persuasion domain and even when provided with only a single short prompt[17,24].

Overall, we believe that the literature offers ample reason to be concerned about the possibility of large-scale political manipulation of people's attitudes and opinions based on exploitation of their personal characteristics and vulnerabilities.

At present, very little is known about how one might counter the effects of psychological microtargeting. One recent study employed a well-known practice from inoculation theory called 'boosting', which involved providing participants with information about their personality before targeted ads were displayed[25]. This intervention proved sufficient to boost participants' ability to detect microtargeted advertisements. However, the study measured only detection skills and not persuasiveness[25], so it is unknown whether detection is sufficient to attenuate the persuasive advantage of targeted messages.

Another approach is to 'reverse-engineer' psychological microtargeting using Large Language Models (LLMs). This method is supported by the development of a computational text model capable of inferring people's personalities based on the texts they consume online[20]. This model has the ability to determine if there is a match between the personality trait of an individual and an ad. If that match is high, it might arguably be the result of

microtargeting, opening the door to potentially warning users that they may be targeted.

In the present investigation, we examine the application of such a model to counter the impacts of microtargeting by employing a warning "popup" triggered when the alignment between a user's personality and a political message seems "too good to be true". Specifically, the language model created by Simchon et al. [20] can be employed, with user consent, to analyze incoming messages. The model generates a warning signal when it detects a personality-based match that may suggest manipulative persuasion. In this report, we present three new studies that explore the effectiveness of this potential countermeasure against microtargeting.

The studies were conducted with participants' prior consent and relied on knowledge of their personalities. This information was then used to identify messages aligning with each participant's personality. The messages were selected from a pool of 1552 political ads published on Facebook for UK users between December 2019 and December 2021. In each study, the focus was on comparing the persuasiveness of targeted messages presented without a warning to those accompanied by a "popup" warning that highlighted potential manipulation. The popup alert was overlaid on the selected targeted ads and remained on the screen until the participant clicked a button. Our hypothesis was that the popup warning would be effective in reducing the persuasiveness of the targeted ads.

## Methods

Our studies depart from the previous finding that personalized political ads tailored to individuals' personalities are judged to be more persuasive than non-personalized ads[17]. That study focused on a specific aspect of personality, namely openness to experience, which a language model had proven particularly adept at predicting based on users' consumed text[20]. For this reason, the ads used in the present experiments also involved openness to experience. Examples of ads classified by the model developed by[20] according to their openness levels are depicted in Fig. 1.

The core manipulation in our studies involved the match (vs. mismatch) between participants' actual personality and the presumed appeal of ads to people with that personality.

### Participants selection

Our samples for Study 1 and Study 2a were drawn from pools of subjects who had participated in prior surveys related to ads persuasiveness and described in[17], in which the openness levels of the participants were assessed using the BFI-2[26]. To ensure that our results were not influenced by participants' familiarity with our previous experiments on microtargeting, we decided to use a sample of individuals for Study 2b who had never been exposed to such studies. In all three cases, we selected individuals from each pool who fell either in the top or bottom 30% of openness scores in order to be able to unambiguously select ads (i.e., low- or high-openness) that targeted their personality. To ensure that only these selected participants were included in our studies, we created an approved list of their IDs when setting up the study on Prolific. Participants in Study 1 ($M_{age} = 44.37$, $SD_{age} = 13.74$) included 385 women, 278 men, 1 non-binary participant, and 2 participants with undisclosed gender. Participants in Study 2a ($M_{age} = 44.34$, $SD_{age} = 12.94$) included 305 women, 121 men, 4 non-binary respondents, and 2 participants with undisclosed gender. Participants in Study 2b ($M_{age} = 41.53$, $SD_{age} = 13.24$) included 413 women, 251

**Fig. 1 | Examples of low- and high-openness advertisements.** Panels (**A**, **B**) illustrate examples of low- and high-openness ads respectively, as classified by the language model developed by[20].

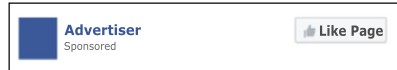

men, 4 non-binary participants, and 1 respondent with undisclosed gender. All studies were fully reviewed and approved by the School of Psychological Science Research Ethics Committee at the University of Bristol (ethics approvals #17566 and #17923). All participants provided informed consent via mouse click prior to their participation. The studies lasted ~8 min, and the participants received £1.20 as compensation.

## Ads selection and classification

Ads were sampled from a pool of 1552 political messages published on Facebook to UK users between December 2019 and December 2021. Openness levels were assigned to each ad using a language model predicting the appeal of the ads' text for the openness to experience personality dimension[20].

## Study design

Our studies are based on a pilot study ($N = 565$) with a between-subjects design, where participants were divided into two groups and equally exposed to personality-targeted ads ($N = 5$) and non-targeted ads (N = 5). In the intervention group, targeted ads were introduced with a popup that warned participants about the potential microtargeting nature of the ad. The warning was not presented to the control group. The results revealed a non-significant reduction of persuasiveness in the intervention group compared to the control. Further details are presented in the Supplementary Note 3.

To verify the robustness of our findings, we conducted post hoc power analyses. Since observed power calculations, where the target effect size comes from the data used, give misleading results, we opted for a specific smallest effect size of interest (SESOI). We agreed to set the SESOI to correspond to a 5% persuasiveness reduction in targeted ads preceded by popups when compared to targeted ads without popups. Applying this would be straightforward in the three main studies, as the regression intercepts represent the average persuasiveness for targeted-no-popup ads when the matching score is at its mean. However, the pilot study was less straightforward, as we compared group estimates that average the persuasiveness of non-targeted and targeted ads (with or without a popup, depending on the group). Therefore, we set our SESOI for the pilot study to a 2.5% reduction of the control group's average persuasiveness (the intercept in our regression model). This represents our hypothesis that popups would reduce the persuasiveness of targeted ads by 5%, which, given that half the ads in each group were targeted and the other half were not, translates to a 2.5% reduction in overall group persuasiveness.

We calculated the power of our studies using the *simr* R package[27], with the number of simulations set to 1000. The results for the pilot study revealed a power of 25.30% (CI = [22.63, 28.11]), suggesting that that study was not sufficiently powered to detect such a small effect with a between-subjects design. Additionally, the pilot study lacked a condition where the same person was exposed to targeted ads both with and without a popup, making it difficult to directly compare the popup's effect on the same individual's response. To address these limitations, we designed Study 1, Study 2a, and 2b with a within-subjects design, where participants were exposed to all three types of ads. Importantly, the three studies proved to be well-powered to detect the SESOI, exhibiting much higher power, at 99.50% (CI = [98.84, 99.84]), and 94.60% (CI = [93.01, 95.92]), and 99.10% (CI = [98.30, 99.59]) respectively.

In the three within-subject studies, participants were exposed to both non-targeted ($N = 5$) and targeted ads ($N = 10$). Within the 10 targeted ads, 5 were randomly chosen for each participant to display the popup. The remaining 5, while still targeted, did not include a popup. In Study 1, the popup also included a brief explanation of psychological microtargeting and its associated dangers. In the subsequent studies, such explanation was moved to the instructions. The popup appeared superimposed on the ad and remained on screen until the participant clicked a close button. We specified that the popup appeared before some, but not all, targeted ads.

## Differences between studies

We developed Study 2a to build on the intervention tested in Study 1 and to increase the likelihood of detecting an effect with a more optimized

approach. Study 2a was conducted 3 months after the completion of Study 1. In this follow-up study, we further enhanced the instructions by stating that the popup was automatically generated by a browser extension which analyzed the ad content, cross-referenced it with participants' responses to a prior personality assessment, and signaled a potential microtargeted ad. We also specified that the browser extension solely analyzed the text of the ads, focusing on textual features without interpreting the intended message of the ads. We excluded advertisements related to governmental recommendations, such as COVID-19 related measures. This decision was made to eliminate potential confusion arising from the warning popup identifying a valid scientific recommendation as a microtargeted advertisement, as could have occurred in Study 1. In addition, the brief explanation of psychological microtargeting was moved from the popup text to the instructions to improve readability. The final version of the popup warning used in Study 2a was phrased as follows: "This advertisement appears to be written in a style tailored to appeal to you. If you encounter such content, it is likely a result of psychological microtargeting." Furthermore, we included a comprehension check after the initial set of instructions. This check involved a randomly selected multiple-choice question from a pool of four options that we created. The purpose of this check was to ensure participants' understanding of the assignment. If participants answered incorrectly, the instructions were presented once again. Further details regarding the study design, such as the instructions and the popup texts, are reported in the Supplementary Notes 1 and 2. In contrast, Study 2b was designed to mirror Study 2a, with the sole difference being that it was conducted on a sample that had never been previously exposed to our microtargeting experiments. Study 2b took place 8 months after the completion of Study 2a.

Study 1 was preregistered at https://aspredicted.org/2je46.pdf on December 14, 2023. Study 2a was not preregistered since it shared the same study design and structure of Study 1, with the exception of refined instructions, popup text, and ads. When preregistering Study 1, we stated the intention to assess the effectiveness of the intervention across various items on our scale. We report the results of this analysis in the Supplementary Note 5. We slightly deviated from our preregistration by mean-centering the matching score and adding random slopes in the regressions. Both deviations were justified by an increase in interpretability, and the improved performance of the revised model compared to the preregistered one. In the Supplementary Note 6, we report the results for the preregistered models and demonstrate that they overlap with those presented here. Finally, Study 2b was preregistered at https://aspredicted.org/fkxw-hq3p.pdf on October 14, 2024.

## Intervention and measurements

In our studies, ads were shown in a Facebook ad format in random order, with the advertiser's name and any identifying information removed. Ads did not display images to ensure participants focused on the text. For each ad, persuasiveness was measured using a 5-point agreement scale, with items adapted from[28] also used in previous related studies[17]. Further details on the scale used can be found in the Supplementary Note 4.

There is an ongoing debate about whether self-reported persuasiveness accurately reflects the actual persuasiveness of a message. Some studies have found a correlational[29] or causal[30] link between perceived and actual message persuasiveness. However, critics contend that individual-level correlations between perceived and actual message persuasiveness do not necessarily licence conclusions about message-level correlations[31,32]. In other words, whether a group of participants perceives a particular message as persuasive may be independent of the message's actual level of persuasiveness. Nonetheless, recent longitudinal randomized studies have provided evidence against the critics' claims by showing that reported persuasiveness does serve as a valid proxy for actual message effectiveness. Specifically, Ma et al.[33] demonstrated that vaping prevention ads perceived as more persuasive at the beginning of their study led to lower vaping susceptibility among participants much later on, and that this effect was fully mediated by the perceived potential of the messages to change behavior (i.e., perceptions of the message). Moreover, in our studies, the perceived persuasiveness

scales employed also include measures of attitudes and intentions (e.g., "Overall, I like this ad," "I would click on this link after seeing the ad"), both of which have been found to be reliable indicators of message-level persuasiveness[34].

## Data analysis

Following[17], we fitted a linear mixed-effects model to predict self-reported persuasiveness with a two-way interaction between "ad type" (a factor with three levels: "non-targeted", "targeted-no popup", and "targeted-popup"), and a personality matching score. This score represents the absolute difference between the scaled openness scores of a $participant_i$ and an $ad_j$. The former were obtained from previous surveys we conducted on ads persuasiveness[17] (with the exception of Study 2b, where we used a "fresh" sample), whereas the latter were calculated using the language model described in[20]. The final matching score was extracted as follows: $Matching_{i_j} = |z(Openness_i) - z(Openness_j)|$. By incorporating the matching score, we took into account each person's degree of openness. We also opted to mean-center the matching score to accommodate its varying ranges across ad types (i.e., non-targeted ads inherently are less likely to match a person's personality). This centering ensures that the mean persuasiveness within each ad type is equal to 0, thus simplifying the interpretation of the regression results. Additionally, we did not average persuasiveness rates within participants. Instead, we specified random intercepts and slopes for both participants and ads to account for the variability in the effects of ad types within each participant and each ad. Participants who provided identical responses to all items were excluded from the data analysis in all three studies.

Data was assumed to follow a normal distribution for the analysis. Post-analysis evaluation of model assumptions through diagnostic plots supported this decision: Q-Q plots showed adequate normality with some tail deviations, while residual plots revealed patterns typical of Likert scale data. While minor violations of normality and homoscedasticity assumptions were observed, these are common with Likert data and linear mixed effects models are robust to such departures[35,36].

## Reporting summary

Further information on research design is available in the Nature Portfolio Reporting Summary linked to this article.

## Results

In Study 1, the results reveal a non-significant effect of the popups on persuasiveness ($t$(59, 940) = -0.729, $p = 0.466$, $\beta = -0.026$, 95% CI = [-0.097, 0.044]). By contrast, non-targeted ads show a significant decreasing effect on persuasiveness ($t$(59, 940) = -2.734, $p = 0.006$, $\beta = -0.215$, 95% CI = [-0.369, -0.061]) when compared to targeted ads with no popups. Taken together, the two outcomes reveal the effectiveness of microtargeting accompanied by the failure of the popup to eliminate that persuasive advantage.

Surprisingly, the centered matching score becomes significant only when considered in interaction with non-targeted ads ($t$(59, 940) = -2.870, $p = 0.004$, $\beta = -0.210$, 95% CI = [-0.354, -0.067]), meaning that the more non-targeted ads differ from participants' levels of openness, the less persuasive they are reported to be. This also implies that targeted ads (with or without a popup) were generally considered as more persuasive than non-targeted ones, regardless of the exact proximity between ads' and participants' levels of openness.

In Study 2a ($N = 432$), we maintained the exact within-subject design of Study 1 but introduced refined instructions, different ads, and a streamlined popup text. We applied the same linear mixed model formula used in Study 1 to the data collected in Study 2a. As indicated in Table 1, the popup did not have a significant effect on self-reported persuasiveness ($t$(38, 880) = -0.897, $p = 0.370$, $\beta = -0.033$, 95% CI = [-0.106, 0.039]). At the same time, non-targeted ads were still associated with a significant decrease in persuasiveness ($t$(38, 880) = -2.112, $p = 0.035$, $\beta = -0.149$, 95% CI = [-0.287, -0.011]).

Differently from Study 1, the persuasive advantage of targeted ads was further confirmed by the significant negative effect of the mean-centered matching score ($t$(38, 880) = -2.012, $p = 0.044$, $\beta = -0.162$, 95% CI = [-0.319, -0.004]). This replicates previous findings[17] and indicates that as the difference between the person's openness and the ad's openness increased (i.e., higher matching score), the targeted ads were perceived as less persuasive.

Furthermore, the interaction between non-targeted ad type and matching score was not statistically significant in Study 2a ($t$(38, 880) = -0.429, $p = 0.668$, $\beta = -0.048$, 95% CI = [-0.268; 0.172]). This also contrasts with the findings from Study 1, where the interaction between non-targeted ads and matching score was negative and significant. This non-significant interaction suggests that for non-targeted ads, the effect of matching score on persuasiveness did not differ compared to targeted ads. In other words, non-targeted ads were generally seen as less persuasive, and this lower persuasiveness did not vary based on the matching score.

In Study 2b ($N = 669$), we applied the same design and analysis methods as in Study 2a, but with a different sample of participants. Interestingly, the popup showed a small but narrowly significant negative effect on persuasiveness ($t$(60, 210) = -1.998, $p = 0.046$, $\beta = -0.063$, 95% CI = [-0.126, -0.001]). Consistent with the findings of the other two studies, non-targeted ads were associated with a significant decrease in persuasiveness ($t$(60, 210) = -2.129, $p = 0.033$, $\beta = -0.171$, 95%CI = [-0.328, -0.014]). However, the centered matching score did not achieve significance, either on its own or in interaction with ad types. Further details of the regressions for the three studies are shown in Table 1 and Fig. 2.

Finally, we decided to conduct non-preregistered equivalence tests for our studies using the *parameters* R package[27]. Equivalence tests are used to determine whether the difference between two conditions is small enough to be considered negligible or "equivalent." In contrast to traditional null hypothesis significance testing (NHST), which aims to detect any difference (no matter how small), equivalence testing seeks to demonstrate that the difference is sufficiently small to be considered practically equivalent[37]. The key idea is to define a region of practical equivalence (ROPE) around the null value (typically 0), and then test whether an estimate and its confidence intervals fall within this ROPE. If they do, one can conclude that the two conditions are equivalent, or practically the same, for the purposes of the study. In all of our studies, the popup effect falls within the bounds of what is practically equivalent to 0 (Study 1: 90% CI = [-0.085, 0.033]; Study 2a: 90% CI = [-0.094, 0.027]; Study 2b: 90% CI = [-0.115, -0.011]). The ROPE for the equivalence tests was set to the default bounds for our models' properties, calculated as $[-0.1*SDy, 0.1*SDy]$. This corresponds to the range [-0.126, 0.126]. This confirms that, although the popup effect reached statistical significance in Study 2b, it remains practically equivalent to the effects of targeted ads without a popup. Further details on the equivalence tests are illustrated in Fig. 3.

## Discussion

The purpose of this investigation was to test the efficacy of a warning popup in countering the persuasiveness advantage of psychologically microtargeted ads. We conducted three within-subject studies, whose results replicate the targeting effect observed in[17]. Specifically, both targeting conditions were significantly more persuasive than the non-targeted ads within the same participants, confirming that personality-based targeting can be effective.

Of greater interest is the finding that the popup failed to effectively counter the microtargeting effect in our studies. While the popup led to a slight decrease in persuasiveness, this decline was statistically discernible only in one of the three studies. More important, the results of our equivalence tests indicate that the effects of the popup warnings in all three studies fall within the predefined region of practical equivalence for all studies, which is why we do not consider the narrowly significant result from Study 2b to warrant further investigation. The equivalence tests imply that the impact of popups on persuasiveness is not practically relevant or meaningful. Targeting an aspect of personality appears to not only enhance persuasiveness but also creates resistance to pre-exposure warnings. In light of contemporary legislative initiatives, such as the EU's AI Act, which place

**Table 1 | Results of Linear Mixed-Effects Model for Studies 1, 2a, and 2b**

| | Dependent variable: | | |
|---|---|---|---|
| | **Persuasiveness** | | |
| | **(Study 1)** | **(Study 2a)** | **(Study 2b)** |
| ad_typepopup | -0.026 | -0.033 | -0.063 |
| | (-0.097, 0.044) | (-0.106, 0.039) | (-0.126, -0.001) |
| | $p = 0.466$ | $p = 0.370$ | $p = 0.046$ |
| ad_typenon_targeted | -0.215 | -0.149 | -0.171 |
| | (-0.369, -0.061) | (-0.287, -0.011) | (-0.328, -0.014) |
| | $p = 0.007$ | $p = 0.035$ | $p = 0.033$ |
| matching_score_centered | 0.036 | -0.162 | -0.091 |
| | (-0.049, 0.120) | (-0.319, -0.004) | (-0.212, 0.031) |
| | $p = 0.407$ | $p = 0.045$ | $p = 0.143$ |
| ad_typepopup:matching_score_centered | 0.052 | 0.163 | -0.002 |
| | (-0.060, 0.164) | (-0.030, 0.356) | (-0.148, 0.143) |
| | $p = 0.365$ | $p = 0.098$ | $p = 0.977$ |
| ad_typenon_targeted:matching_score_centered | -0.210 | -0.048 | -0.139 |
| | (-0.354, -0.067) | (-0.268, 0.172) | (-0.310, 0.032) |
| | $p = 0.005$ | $p = .668$ | $p = 0.112$ |
| Constant | 2.690 | 2.748 | 2.832 |
| | (2.570, 2.809) | (2.624, 2.871) | (2.704, 2.959) |
| | $p < 0.001$ | $p < 0.001$ | $p < 0.001$ |
| Observations | 59,940 | 38,880 | 60,210 |
| Log Likelihood | -86,243.700 | -57,251.780 | -89,003.000 |
| Akaike Inf. Crit. | 172,525.400 | 114,541.600 | 178,044.000 |
| Bayesian Inf. Crit. | 172,696.400 | 114,704.400 | 178,215.100 |

Results of the generalized linear mixed-effects model used in Study 1 (left), Study 2a (center), and Study 2b (right), with self-reported persuasiveness as dependent variable. Estimates are provided together with their 95% confidence intervals. In all models, the observations (i.e., the persuasiveness ratings) are reported individually rather than being averaged within subjects. To address the fact that each participant rated multiple ads on various scale items, and that the same ads appeared under different conditions, random slopes for ad type were included for both participants and ads. Additional variables considered in the model include ad type (with 'targeted-no popup' used as reference level) and the matching score, which has been mean-centered within each ad type and represents the scaled difference between the participant's openness score and that of the ad. Both ad type and the matching score are included in the formula as a two-way interaction.

considerable emphasis on transparency as a regulatory tool, the observed absence of an effect linked to a transparency measure raises potential concerns.

## Limitations
The reasons why the intervention did not reduce the effects of psychological microtargeting may arise from limitations of our study. Research on transparency demonstrates that individuals are generally less willing to accept the use of their personal information for targeting when this information is obtained through cross-website tracking or inferred by the platform based on their behavior, rather than explicitly provided by them[38]. Participants tend to view these practices as intrusive and are more likely to deem such advertising practices as unacceptable. Our study's instructions, on the other hand, made it clear that the information collected to target the ads came from the same platform used for the surveys (i.e., Prolific). Additionally, we explicitly stated that participants' responses to previous surveys were used to ascertain the personality traits being targeted, eliminating any covert inference of these traits by us. Collectively, these factors may render the microtargeting in our experiment more acceptable to participants, thereby diminishing the impact of a popup warning.

Another reason why a popup warning may fail to decrease the effects of microtargeting may lie in the so-called "acceptability gap"[8]. The acceptability gap refers to the finding that individuals perceive personalized content (e.g., customized search results, online advertising, entertainment recommendations) as more acceptable than the collection and use of their personal data for such personalization, even though this personal data is required to provide the personalized services. This suggests that people value

the benefits of personalized services, but underestimate or are unaware of the data collection required to deliver those services. In our case. this discrepancy may lead individuals to ignore or downplay the significance of popup warnings, as they prioritize the convenience and relevance of personalized content over concerns about data privacy.

The limited effectiveness of the warning popups in our studies raises questions about the efficacy of transparency measures. Our findings align with a broader pattern observed across various domains where transparency or warnings have shown limited impact. For instance, the misinformation literature has consistently demonstrated a "continued-influence effect", where corrected information continues to influence people's beliefs and reasoning, even when they acknowledge the correction[39,40]. Similarly, research has shown that explicit warnings about potential misinformation, whether general or specific, can reduce but not eliminate reliance on false information[41]. This persistence of influence extends to fictional contexts as well, where readers tend to rely on information from stories even when it contradicts known facts, and warnings about potential errors in fiction have limited effect in reducing suggestibility[42]. These parallels suggest that the challenge of countering the effects of microtargeting through transparency measures may be part of a more general phenomenon where human cognition struggles to fully discount or disregard information once it has been processed, regardless of subsequent warnings or corrections.

## Conclusions
Our results have important implications for online transparency measures, including those used by social media platforms that implement prompts or

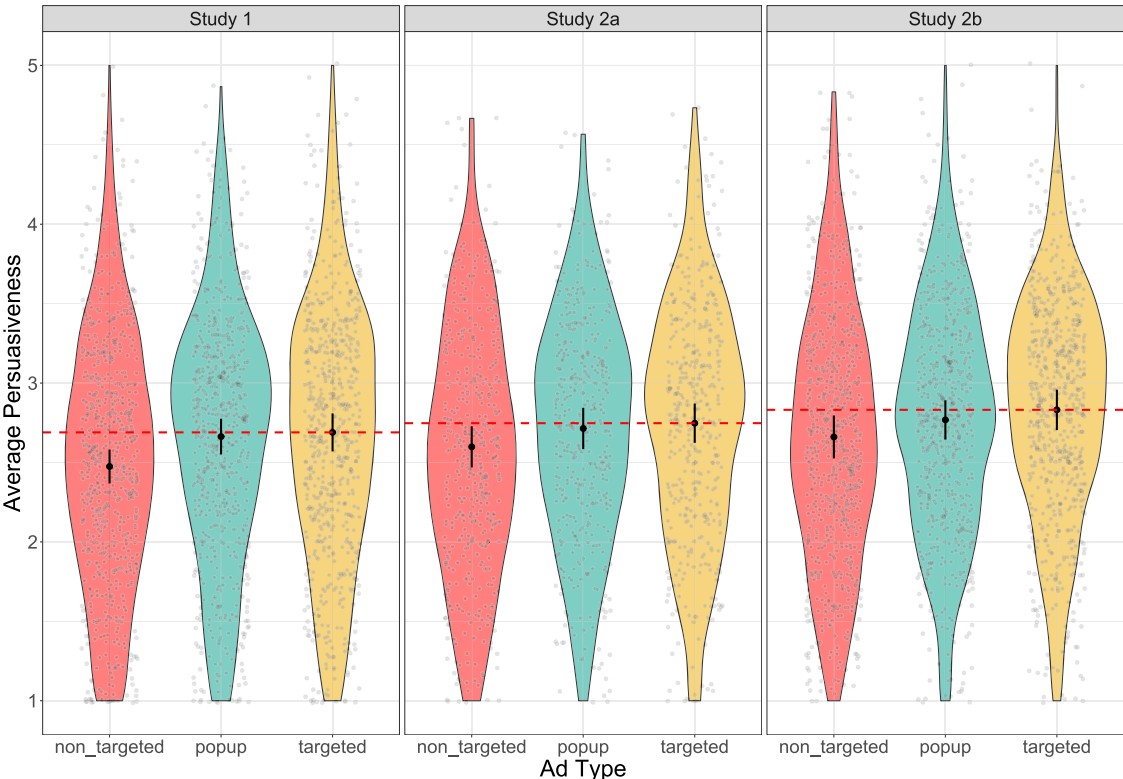

**Fig. 2 | Distributions of average persuasiveness scores across conditions in studies 1, 2a, and 2b.** Distributions of average persuasiveness scores within subjects across conditions in Study 1 (left, $N = 666$), Study 2a (center, $N = 432$), and Study 2b (right, $N = 669$). In each study, each participant provided ratings for three groups of ads: non-targeted ads (red), targeted ads without a warning popup (yellow), and targeted ads with a warning popup (blue). The point ranges within the violin plots depict the 95% confidence intervals for the estimated marginal means of persuasiveness for each ad type. These values were extracted from the regression models and represent the average estimated persuasiveness for each level of the "ad type" predictor. In Study 1, the marginal means of persuasiveness are $M_{non-targeted} = 2.47$, $M_{popup} = 2.66$, and $M_{targeted} = 2.68$. The patterns are similar in Study 2a, with $M_{non-targeted} = 2.59$, $M_{popup} = 2.71$, and $M_{targeted} = 2.74$, as well as in Study 2b, with $M_{non-targeted} = 2.65$, $M_{popup} = 2.76$, and $M_{targeted} = 2.84$. In each panel, the red dotted line represents the marginal mean of reference (i.e., the targeted ads without a popup).

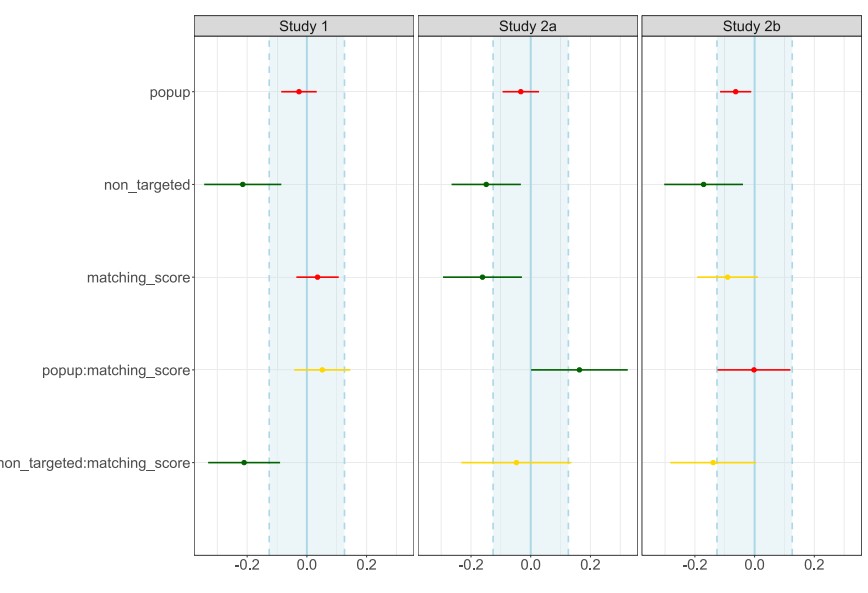

**Fig. 3 | Equivalence test results for studies 1, 2a, and 2b.** Results for the equivalence tests of Study 1 (left), Study 2a (center), and Study 2b (right). The tests are calculated based on the linear mixed-effects models described in the Methods section, and include the effects of ad type (popup and non-targeted), the centered matching score, and the fully-crossed interaction between the two. The point ranges for each variable correspond to the model estimate and its 90% narrow confidence intervals. The light blue area designates the region of practical equivalence (ROPE), which in these two cases corresponds to [-0.126, 0.126]. Practical equivalence is accepted (in red) when the narrow confidence intervals are completely within the ROPE, regardless of statistical significance. Practical equivalence is instead rejected (in green) when the coefficient is statistically significant, whether the narrow confidence intervals (i.e., $1 - 2*\alpha$) include or exclude the ROPE boundaries, as long as the narrow confidence intervals are not fully covered by the ROPE. Finally, the decision to accept or reject practical equivalence is undecided (in yellow) when effects are not statistically significant and the narrow confidence intervals overlap the ROPE.

warnings similar to the interventions tested in our studies. For instance, features like X/Twitter's Community Notes or Facebook's fact-check tags aim to provide users with additional context or transparency regarding certain content. While these interventions focus more on combating misinformation rather than microtargeting, and are directed at all users rather than shown to specific individuals, their effectiveness appears to be similar to what we found in our results. Specifically, they either show no evidence of improvement in online behavior[43] or have minimal impact on user beliefs, depending on several potential limitations, such as platform enforcement, the clarity of warnings, and possible backfire effects[44,45]. In our studies, the latter case may have occurred, with participants interpreting the warnings as confirmation that the advertised content was well-suited to their personalities, therefore reporting levels of persuasiveness equal to targeted ads without warnings.

However, it is also important to note that our findings may not necessarily indicate a lack of participants' concern about microtargeting, but rather highlight the complexity of individuals' reactions to it. These factors include their awareness and concern about the message being microtargeted, their perception of its persuasiveness, and whether they ultimately feel persuaded by it. As a result, our findings highlight the importance of conducting a more comprehensive assessment of these elements in order to better understand the implications of microtargeting practices and inform strategies for user data protection and transparency.

A final aspect to consider is that the popups may not be sufficiently effective in reducing perceived persuasiveness but might be effective in eliminating or reducing different aspects of a targeting event. Participants might still perceive targeted ads as persuasive, recognizing them as well-tailored to their preferences—even in presence, or perhaps because of, a warning. However, the positive impact of the popup might manifest differently, for example in diminishing interaction with the displayed ads—whether through clicking for more information or sharing to amplify their reach. Additionally, the warnings may increase users' awareness of the technology behind microtargeting, potentially highlighting the harm it causes and prompting second-order effects, such as regulatory action or the promotion of more transparent design choices.

Future research should explore these implications in greater depth. To this regard, we are planning a series of upcoming studies to investigate the (in)effectiveness of microtargeting based on different personality traits, such as extraversion. Furthermore, given that individuals' willingness to be microtargeted varies by ad type[5,6], we also plan to assess the effects of microtargeting across different ad categories (e.g., beauty products). Subsequent studies should examine how explicitly referencing large language models (LLMs) in pop-up notifications influences user perceptions and trust, as this could significantly alter their willingness to engage with microtargeted content. Lastly, future efforts should explore the combined impact of multiple interventions[46], such as using warning popups together with boosting strategies[25], to maximize the effectiveness of these measures and enhance users' autonomy.

## Data availability
The data for this manuscript are stored on OSF under accession code https://doi.org/10.17605/OSF.IO/RXHTC[47].

## Code availability
The scripts to reproduce the results and the graphs presented in this manuscript are available on OSF under accession code https://doi.org/10.17605/OSF.IO/RXHTC[47]. The repository includes guidance on how to use the code.

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

## Acknowledgements

This research was supported by the Volkswagen Foundation (grant "Reclaiming individual autonomy and democratic discourse online: How to rebalance human and algorithmic decision making"). SL also acknowledges financial support from the European Research Council (ERC Advanced Grant 101020961 PRODEMINFO), the Humboldt Foundation through a research award, and the European Commission (Horizon 2020 grant 101094752 SoMe4Dem). For the purpose of open access, the author(s) has applied a Creative Commons Attribution (CC BY) licence to any Author Accepted Manuscript version arising from this submission. The funders had no role in study design, data collection and analysis, decision to publish or preparation of the manuscript. The authors thank Dr. Mattan Ben-Shachar for his assistance.

## Author contributions

F.C., A.S., M.E., and S.L. conceptualized the research. F.C. performed the data collection and analysis. F.C. and S.L. wrote the original draft of the manuscript.

## Competing interests

The authors declare no competing interest.
