## [Transparent Peer Review file · Communications Psychology]

Warning people that they are being microtargeted fails to eliminate persuasive advantage

Corresponding Author: Dr Fabio Carrella

Version 0:

Decision Letter:

Dear Dr Carrella,

Thank you for submitting your manuscript titled "Transparency is not enough: Warning people that they are being microtargeted fails to eliminate persuasive advantage" to Communications Psychology. We have given the paper our careful consideration and find it of potential interest. However, due to certain shortcomings we are concerned that sending the current manuscript out to review could lead to unnecessary delays and quite possibly an undesirable outcome of the review process.

In particular, for manuscripts that report null results, we require the following:

- Evidence that the study is sufficiently powered to detect the smallest theoretically or pragmatically meaningful effect
- Bayes Factors or equivalence tests to interpret the null results
- Appropriate language to describe the results.

We would therefore like to invite you to revise your manuscript to address these concerns before we make a final determination on whether to send your manuscript for external review.

We shall hope to receive your revised version as soon as you are able to complete the suggested revisions. If something similar is published in the interim we will have to consider the impact it has on the novelty of a revised manuscript.

If you anticipate a delay of more than four weeks, please let us know. Should your manuscript be substantially delayed without notifying us in advance and your article is eventually published, the received date may be that of the revised, not the original, version.

We also ask that you ensure your manuscript complies with our editorial policies and reporting requirements.

To that end, we require revised manuscripts to be accompanied by two completed items: a reporting summary that collects information on study design and procedure, and an editorial policy checklist that verifies compliance with all required editorial policies.

- <https://www.nature.com/documents/nr-reporting-summary.zip>>Nature Research Reporting Summary
- <https://www.nature.com/documents/nr-editorial-policy-checklist.pdf>>Editorial Policy Checklist

All points on the policy checklist must be addressed. Your revised manuscript can only be sent to referees if these checklists are completed and uploaded with the revision.

If you are not interested in submitting a suitably revised manuscript in the future please let me know immediately so we can close your file. If you have any questions, please contact me.

Please use the link below when you are prepared to resubmit.
Link Redacted

Thank you for your interest in Communications Psychology.

Best regards,
Samantha Antusch

Samantha Antusch, PhD

Senior Editor
Nature Human Behaviour

Consulting Editor
Communications Psychology

Version 1:

Decision Letter:

Dear Dr Carrella,

Thank you for your patience during the peer-review process. Your manuscript titled "Transparency is not enough: Warning people that they are being microtargeted fails to eliminate persuasive advantage" has now been seen by 2 reviewers, whose comments are appended below. You will see that they find your work of some potential interest. However, they have raised quite substantial concerns that must be addressed. In light of these comments, we cannot accept the manuscript for publication, but would be interested in considering a revised version that fully addresses these serious concerns.

We hope you will find the Reviewers' comments useful as you decide how to proceed. Should additional work allow you to address these criticisms, we would be happy to look at a substantially revised manuscript. If you choose to take up this option, please highlight all changes in the manuscript text file, and provide a detailed point-by-point reply to the reviewers.

Editorially, we request that you generalize your findings to a new sample that is not known to have already participated in studies on misinformation/persuasion interventions. Additionally, please expand the discussion to include the limitations of the study noted by the reviewers as well as better integrating the current findings into the existing literature and consideration of their implications. Please provide the additional details regarding the methods.

I am attaching a checklist that details critical reporting requirements for the revised manuscript. Please attend to each item and ensure your manuscript is fully compliant. We are requesting that your manuscript aligns with these requirements as this facilitates the evaluation of your manuscript, reducing delays in re-review and potential future acceptance. If your revised manuscript is not aligned with these requests on major issues, such as those concerning statistics, it may be returned to you for further revisions without re-review. Additional information can be found in our style and formatting guide Communications Psychology formatting guide.

If the revision process takes significantly longer than five months, we will be happy to reconsider your paper at a later date, provided it still presents a significant contribution to the literature at that stage.

Please use the following link to submit your

- revised manuscript,
- point-by-point response to the referees' comments,
- cover letter (as a separate document),
- the Editorial Policy Checklist (see below),
- the Reporting Summary (see below), and
- the completed Editorial Request Table (attached):

Link Redacted

Thank you for the opportunity to review your work.

Best regards,

Jennifer Bellingtier

Jennifer Bellingtier, PhD
Senior Editor
Communications Psychology

REVIEWER EXPERTISE:

Reviewer #1 persuasion

Reviewer #2 persuasion

REVIEWER REPORTS:

Reviewer #1 (Remarks to the Author):

I would like to thank the authors for the opportunity to read the submitted manuscript. The topic is highly relevant, and I'm a big fan of null results given an underlying high quality approach. In this respect, I very much appreciate the extensive open materials, code, etc.; this is textbook done!

In general, the tested idea and especially its practical design provide interesting & valuable insights, the study designs make sense, and the analyses are sound. That said, there are still a few issues that need to be addressed.

Major revisions:

1. I'm a bit confused by the fact that trials 1 and 2 seem to be running at exactly the same time (if I understand correctly)? To me it reads as if study 1 was pre-registered and conducted first and then study 2 was planned with the idea of maybe finding an effect with an optimised intervention. I think it would be more logical to present the trials in a more sequential way, i.e. to present trial 2 as a further development of the intervention tested in trial 1. However, I understand that the manuscript is very limited in space. Perhaps the authors could at least include a clarifying outline of the timing of the studies in the Supplementary Materials.
2. The authors discuss a number of limitations, which I really appreciate. However, the pre-selected sample of participants - i.e. they had already participated in studies on misinformation/persuasion interventions - is not addressed. In my opinion, this is the main reason why the authors do not find significant effects of the intervention. The null results may simply be the result of a group of participants who are more accustomed to intervention experiments, especially ones they have done in the immediate past. This limitation needs to be discussed.
3. Also regarding this comment (p. 11, line 326): "Collectively, these factors may render the microtargeting in our experiment more acceptable to participants" → I thought this could go even further. That is, the participants might also have understood the warnings as a confirmation of how well the advertised content really suited them/their personality. In this case, they might have been more wary of the technology behind it, but still interested/convinced because of the perceived higher personal fit.
4. I missed practical implications in the discussion section. The authors start this section with a brief interpretation of the present results, but then focus on the limitations of the results. I think it would be important to outline what the present results say about social media platforms that use such or similar prompts (see e.g. X/Twitter's community notes), how the interventions used are comparable or different, and in which cases or with which modifications the tested intervention might be effective.
5. The same was true for suggestions for future research. Although the authors added a call for next research steps to the various limitations they outlined, it would be interesting (especially for other researchers building on these null results) to describe more concretely which next steps (e.g. studies or intervention designs) should be focused on.

Minor revisions:

1. The most compact recommendation has to do with the organisation of the different sections of the manuscript. Especially for the Introduction, Methods and Discussion subtitles are needed to guide the reader. For example, in the Methods section, "Sample", "Measures and design", "Differences between studies", and "Analytic strategy" could capture the different subsections in a meaningful way.
2. I cannot find references to the Supplementary Materials in the main manuscript. It would be useful to refer to them when discussing the pilot study as well as the main studies 1 and 2.

After addressing the above outlined clarity issues in the manuscript and especially developing the Discussion section further, the manuscript would, in my opinion, be a valuable contribution to Communication Psychology.

Reviewer #2 (Remarks to the Author):

This paper examines the effects of pop-up warnings on micro-targeting ads. Through two within-subject experiments, this study found that micro-targeting ads are more persuasive than those without micro-targeting, and that warnings did not significantly impact the persuasiveness of micro-targeting ads. Although the paper is interesting, I have some main concerns about it, listed below.

1) Rationale for within-subject design

It was hard for me to understand why the researchers selected a within-subject design for the actual studies. Is it because the between-subject design was underpowered? Could this be a sufficient rationale? I believe the within-subject design further reduced the effects of the pop-ups, as it annoyed participants repeatedly exposed to ads.

2) Micro-targeting by matching openness

I am not sure that matching personality based solely on openness is well rationalized. If the study aimed to use the rationale that "LLM has proven particularly adept at predicting based on users' consumed text," I believe your micro-targeting pop-ups should have focused on micro-targeting ads utilizing a large language artificial intelligence model. I believe that if you had included a discussion about how AI powers the micro-targeting ads, the experimental results might have been different.

3) Implication of the study

So, what is the larger implication of this study that contributes to communication psychology? In the current version, I do not see a significant contribution of this study to Communications Psychology.

EDITORIAL POLICIES

We ask that you ensure your manuscript complies with our editorial policies and reporting requirements.

To that end, we require revised manuscripts to be accompanied by two completed items: a reporting summary that collects information on study design and procedure, and an editorial policy checklist that verifies compliance with all required editorial policies

- <https://www.nature.com/documents/nr-reporting-summary.zip>>Nature Research Reporting Summary
- <https://www.nature.com/documents/nr-editorial-policy-checklist.pdf>>Editorial Policy Checklist

All points on the policy checklist must be addressed. Your revised manuscript can only be sent back to the referees if these checklists are completed and uploaded with the revision.

Notes: If you have submitted a Stage 1 Registered Report, Review, Primer, Comment, or Perspective you do not need to submit these forms. If you have already submitted these forms, you may disregard this request.

** Visit Nature Research's author and referees' website at <http://www.nature.com/authors>>www.nature.com/authors for information about policies, services and author benefits**

Version 2:

Decision Letter:

Dear Dr Carrella,

Your manuscript titled "Transparency is not enough: Warning people that they are being microtargeted fails to eliminate persuasive advantage" has now been seen by our reviewers, whose comments appear below. In light of their advice I am delighted to say that we are happy, in principle, to publish a suitably revised version in *Communications Psychology*.

We therefore invite you to revise your paper one last time to address the remaining concerns of our reviewers and a list of editorial requests. At the same time we ask that you edit your manuscript to comply with our format requirements and to maximise the accessibility and therefore the impact of your work.

EDITORIAL REQUESTS:

Please review our specific editorial comments and requests regarding your manuscript in the attached "Editorial Requests Table". In particular, please ensure policy recommendations stay close to what was examined in your study. Please outline your response to each request in the right hand column. Please upload the completed table with your manuscript files as a Related Manuscript file.

SUBMISSION INFORMATION:

OPEN ACCESS:

*** TRANSPARENT PEER REVIEW:** *Communications Psychology* uses a transparent peer review system. On author request, confidential information and data can be removed from the published reviewer reports and rebuttal letters prior to publication. If you are concerned about the release of confidential data, please let us know specifically what information you would like to have removed. Please note that we cannot incorporate redactions for any other reasons.

Link Redacted

Best regards,

Jennifer Bellingtier

Jennifer Bellingtier, PhD
Senior Editor

REVIEWERS' EXPERTISE:

Reviewer #1 persuasion

Reviewer #2 persuasion

REVIEWERS' COMMENTS:

Reviewer #1 (Remarks to the Author):

Given that the authors have been extremely responsive to all comments and suggestions (I especially appreciate the new study responding to the concern of a selective sample & discussing its new results), I have nothing to specifically add and would suggest to accept this manuscript.

Thank you very much, again, for being able to read this manuscript.

Reviewer #2 (Remarks to the Author):

I appreciate the authors' efforts in addressing the concerns I raised regarding the manuscript. The revisions demonstrate a thoughtful engagement with the feedback, and the manuscript has improved. However, I still have some additional thoughts that could enhance the work further.

While the authors mention the dangers of microtargeting driven by large language models (LLMs), the manuscript's manipulation of pop-ups does not explicitly emphasize the LLM aspect and instead broadly addresses psychological microtargeting. For instance, rather than stating the pop-up as, 'This advertisement appears to be written in a style tailored to appeal to you. If you come across such content, it's probably due to psychological microtargeting,' framing it as, 'This advertisement appears to be written in a style tailored to appeal to you. If you come across such content, it's probably due to psychological microtargeting by a Large Language Model, such as GPT,' which could significantly alter viewers' perceptions. Highlighting this distinction and exploring how explicitly referencing LLMs might impact user reluctance and trust would enrich the manuscript's discussion.

Additionally, while the abstract states that 'given the focus on transparency in initiatives like the EU's AI Act, our finding that warnings have little effect has potential policy implications,' I find the explicit policy implications to be underdeveloped in the discussion section. Expanding on how the findings relate to policy frameworks such as the EU's AI Act or other transparency initiatives would strengthen the manuscript's contribution to debates on the regulation and governance of AI-driven technologies.

Response to reviewers

Manuscript COMMSPSYCHOL-24-0292A

We thank the reviewers for their insightful comments that helped us to improve the quality of the paper.

In the following, we respond to reviewers by providing the original reviewer's comments in *italic* text and our respective answers in **bold** font. Text additions to the manuscript are indicated by their corresponding line or section numbers in the main paper.

Reviewer #1:

I would like to thank the authors for the opportunity to read the submitted manuscript. The topic is highly relevant, and I'm a big fan of null results given an underlying high quality approach. In this respect, I very much appreciate the extensive open materials, code, etc.; this is textbook done!

In general, the tested idea and especially its practical design provide interesting & valuable insights, the study designs make sense, and the analyses are sound. That said, there are still a few issues that need to be addressed.

We sincerely thank the reviewer for their kind remarks and their detailed comments on our manuscript. Their insights were very valuable in identifying its weaknesses and suggesting improvements to enhance the overall quality of the paper.

Major revisions:

1. I'm a bit confused by the fact that trials 1 and 2 seem to be running at exactly the same time (if I understand correctly)? To me it reads as if study 1 was pre-registered and conducted first and then study 2 was planned with the idea of maybe finding an effect with an optimised intervention. I think it would be more logical to present the trials in a more sequential way, i.e. to present trial 2 as a further development of the intervention tested in trial 1. However, I understand that the manuscript is very limited in space. Perhaps the authors could at least include a clarifying outline of the timing of the studies in the Supplementary Materials.

We have expanded on the timing of the studies in Section 2.4 (lines 208 and 232). Specifically, we noted that Study 2a was designed to increase the likelihood of finding a significant effect with a more optimized approach and was conducted three months after Study 1. Study 2b (more on that below), which applied the same design as Study 2a to a "fresh" sample, was conducted eight months after Study 2a.

2. The authors discuss a number of limitations, which I really appreciate. However, the pre-selected sample of participants - i.e. they had already participated in studies on misinformation/persuasion interventions - is not addressed. In my opinion, this is the main reason why the authors do not find significant effects of the intervention. The null results may simply be the result of a group of participants who are more accustomed to intervention experiments, especially ones they have done in the immediate past. This limitation needs to be discussed.

We have added a new study, Study 2b (N = 669), which replicates the design of Study 2a but with a completely fresh sample. In this case, we explicitly excluded anyone who had participated in any of our previous surveys on microtargeting and related interventions. The results show a small but significant effect of the popup in reducing persuasiveness; however, equivalence tests indicate that this effect is practically comparable to displaying targeted ads without a popup, consistent with the findings of Study 1 and Study 2a.

3. Also regarding this comment (p. 11, line 326): "Collectively, these factors may render the microtargeting in our experiment more acceptable to participants" → I thought this could go even further. That is, the participants might also have understood the warnings as a confirmation of how well the advertised content really suited them/their personality. In this case, they might have been more wary of the technology behind it, but still interested/convinced because of the perceived higher personal fit.

We now address this possibility in the discussion (Section 4.2). While we cannot rule out a potential "backfiring" effect—where popups may serve as confirmation of targeting and thus increase perceived persuasiveness—we also consider that raising awareness of such practices might be enough to lead to second-order effects, such as regulatory action or the encouragement of more transparent design choices.

4. I missed practical implications in the discussion section. The authors start this section with a brief interpretation of the present results, but then focus on the limitations of the results. I think it would be important to outline what the present results say about social media platforms that use such or similar prompts (see e.g. X/Twitter's community notes), how the interventions used are comparable or different, and in which cases or with which modifications the tested intervention might be effective.

In Section 4.2, we now relate our findings to research on similar prompts used on social media platforms, such as Twitter's community notes and Facebook's fact-check tags. Although our intervention differs from these prompts in certain respects, the results show comparable levels of (in)efficacy. We also discuss potential approaches to enhance the effectiveness of our intervention in the future directions section (see next point).

5. The same was true for suggestions for future research. Although the authors added a call for next research steps to the various limitations they outlined, it would be interesting (especially for other researchers building on these null results) to describe more concretely which next steps (e.g. studies or intervention designs) should be focused on.

We now explicitly reference future studies we are planning. To make our findings more comprehensive, we aim to explore whether the popups are effective with ads where microtargeting is generally more accepted (e.g., beauty products) and examine outcomes when targeting different personality traits. We also suggest that future research could enhance the impact of warning popups by pairing them with other interventions, such as boosting techniques that encourage users to reflect on their own personality attributes.

Minor revisions:

1. The most compact recommendation has to do with the organisation of the different sections of the manuscript. Especially for the Introduction, Methods and Discussion subtitles are needed to guide the

reader. For example, in the Methods section, "Sample", "Measures and design", "Differences between studies", and "Analytic strategy" could capture the different subsections in a meaningful way.

We have added subsections to the Introduction, Methods, and Discussion sections.

2. I cannot find references to the Supplementary Materials in the main manuscript. It would be useful to refer to them when discussing the pilot study as well as the main studies 1 and 2.

We now provide explicit references to the Supplement where necessary (see lines 198, 229, 458, and 462).

After addressing the above outlined clarity issues in the manuscript and especially developing the Discussion section further, the manuscript would, in my opinion, be a valuable contribution to Communication Psychology.

Reviewer #2 (Remarks to the Author):

This paper examines the effects of pop-up warnings on micro-targeting ads. Through two within-subject experiments, this study found that micro-targeting ads are more persuasive than those without micro-targeting, and that warnings did not significantly impact the persuasiveness of micro-targeting ads. Although the paper is interesting, I have some main concerns about it, listed below.

We thank the reviewer for their thoughtful evaluation of the paper and for highlighting key areas of interest and concern.

1) Rationale for within-subject design

It was hard for me to understand why the researchers selected a within-subject design for the actual studies. Is it because the between-subject design was underpowered? Could this be a sufficient rationale? I believe the within-subject design further reduced the effects of the pop-ups, as it annoyed participants repeatedly exposed to ads.

An important reason for conducting follow-up studies with a within-subjects design, which is now made explicit on line 191, is that the previous between-subjects design lacked a condition where the same individual was exposed to targeted ads both with and without popups. Additionally, participants in the within-subjects design were exposed to only five more ads compared to the between-subjects design, totaling 15 ads. We believe this number is sufficient to test our intervention without making the study overly repetitive.

2) Micro-targeting by matching openness

I am not sure that matching personality based solely on openness is well rationalized. If the study aimed to use the rationale that "LLM has proven particularly adept at predicting based on users' consumed text," I believe your micro-targeting pop-ups should have focused on micro-targeting ads utilizing a large language artificial intelligence model. I believe that if you had included a discussion about how AI powers the micro-targeting ads, the experimental results might have been different.

Thank you for your feedback. In our study, we indeed drew upon our previous work (Simchon et al., 2024), where we calibrated the large language model to predict openness in a manner aligned

with ChatGPT's approach to creating micro-targeted ads. Particularly relevant here is a robustness check we conducted: we asked ChatGPT to rewrite almost 1,000 ads to appeal specifically to high and low openness. We then applied our predictive openness model to each variation, demonstrating strong alignment between the rewritten ads and our model's predictions ($p < .001$, Cohen's $d = 0.51$). This calibration supports the model's efficacy in generating text that resonates with individuals scoring high in openness. Additionally, our findings demonstrate that the persuasive effects of real ads tailored to openness and those generated by ChatGPT to appeal to openness are both significant and in the same direction. Therefore, we believe that our previous work should mitigate any concerns related to parallels between real ads tested for microtargeting via LLM and AI-generated ads.

3) Implication of the study

So, what is the larger implication of this study that contributes to communication psychology? In the current version, I do not see a significant contribution of this study to Communications Psychology.

We have enhanced our Discussion section by introducing new connections to similar interventions on social media and their (in)efficacy, as well as discussing further implications of our studies and future research directions. We hope that these additions will strengthen our paper's contribution to the field.

Response to reviewers

Manuscript COMMSPSYCHOL-24-0292A

We appreciate the reviewers for their valuable feedback, which has significantly enhanced the quality of this final draft.

Below, we respond to reviewers by providing the original reviewer's comments in *italic* text and our respective answers in **bold** font.

Reviewer #1:

Given that the authors have been extremely responsive to all comments and suggestions (I especially appreciate the new study responding to the concern of a selective sample & discussing its new results), I have nothing to specifically add and would suggest to accept this manuscript.

Thank you very much, again, for being able to read this manuscript.

We thank the reviewer for their positive feedback and are pleased to know that we effectively addressed the reviewer's previous concerns.

Reviewer #2:

I appreciate the authors' efforts in addressing the concerns I raised regarding the manuscript. The revisions demonstrate a thoughtful engagement with the feedback, and the manuscript has improved. However, I still have some additional thoughts that could enhance the work further.

We thank the reviewer for the positive feedback.

While the authors mention the dangers of microtargeting driven by large language models (LLMs), the manuscript's manipulation of pop-ups does not explicitly emphasize the LLM aspect and instead broadly addresses psychological microtargeting. For instance, rather than stating the pop-up as, 'This advertisement appears to be written in a style tailored to appeal to you. If you come across such content, it's probably due to psychological microtargeting,' framing it as, 'This advertisement appears to be written in a style tailored to appeal to you. If you come across such content, it's probably due to psychological microtargeting by a Large Language Model, such as GPT,' which could significantly alter viewers' perceptions. Highlighting this distinction and exploring how explicitly referencing LLMs might impact user reluctance and trust would enrich the manuscript's discussion.

We agree that the effects of warnings specifically mentioning targeting made by LLMs or AI should be tested, and we have included the reviewer's suggestion under future recommendations. Specifically, we added the following:

Subsequent studies should examine how explicitly referencing large language models (LLMs) in pop-up notifications influences user perceptions and trust, as this could significantly alter their willingness to engage with microtargeted content.

Additionally, while the abstract states that 'given the focus on transparency in initiatives like the EU's AI Act, our finding that warnings have little effect has potential policy implications,' I find the explicit policy implications to be underdeveloped in the discussion section. Expanding on how the findings relate to policy frameworks such as the EU's AI Act or other transparency initiatives would strengthen the manuscript's contribution to debates on the regulation and governance of AI-driven technologies.

We appreciate the reviewer's insightful suggestion regarding the expansion of policy implications in the discussion section. However, in light of the editor's guidance, we will focus on maintaining recommendations that closely align with the study's findings without delving further into the policy implications.